# SpaceServe: Spatial Multiplexing of Complementary Encoders and Decoders for Multimodal LLMs

Zhicheng Li[1,2]   Shuoming Zhang[1,2]   Jiacheng Zhao[1,2]*   Siqi Li[4]   Xiyu Shi[1,2]
Yangyu Zhang[1,2]   Shuaijiang Li[1,2]   Donglin Yu[5]   Zheming Yang[1,2]   Yuan Wen[3]
Huimin Cui[1,2,6]

[1] State Key Lab of Processors, Institute of Computing Technology, Chinese Academy of Sciences, China
[2] University of the Chinese Academy of Sciences
[3] University of Aberdeen   [4] Beijing University of Technology
[5] The University of Illinois Urbana-Champaign   [6] XCORESIGMA CO.,LTD.

## Abstract

Recent multimodal large language models (MLLMs) marry modality-specific vision or audio *encoders* with a shared text *decoder*. While the encoder is compute-intensive but memory-light, the decoder is the opposite, yet state-of-the-art serving stacks still *time-multiplex* these complementary kernels, idling SMs or HBM in turn. We introduce SpaceServe, a serving system that *space-multiplexes* MLLMs: it decouples all modality encoders from the decoder, and co-locates them on the same GPU using fine-grained SM partitioning available in modern runtimes. A cost-model-guided *Space-Inference Scheduler* (SIS) dynamically assigns SM slices, while a *Time-Windowed Shortest-Remaining-First* (TWSRFT) policy batches encoder requests to minimise completion latency and smooth decoder arrivals. Evaluation shows that SpaceServe reduces time-per-output-token by **4.81×** **on average** and up to **28.9×** on Nvidia A100 GPUs. SpaceServe is available at https://github.com/gofreelee/SpaceServe

## 1   Introduction

Multimodal large language models (MLLMs), for example Qwen2-VL[36] and other recent systems[36, 3, 33, 27], have graduated from lab demos to production services that field image-grounded questions, interpret charts, and reason over audio or video snippets. Conceptually, a MLLM marries the unprecedented linguistic capability of a large text-only foundation model with a set of modality-specific encoders that translate pixels, waveforms, or frames into the token space the decoder already understands. This "encoder + shared-text-decoder" blueprint is now standard across state-of-the-art models[3, 33, 18, 36].

As foundation-model decoders continue to improve "for free" (e.g., via larger pre-training corpora [21, 32, 17, 34, 9] or better post-training [26, 25, 28]), attention is shifting toward the encoders. These front-end components are becoming both more powerful and more complex [33, 36, 3, 27]—handling higher-resolution images, longer audio clips, and even variable-length video. Consequently, an MLLM is effectively two decoupled yet complementary neural networks: a compute-intensive, memory-light encoder paired with a memory-hungry, compute-light decoder. This pronounced resource asymmetry remains largely unaddressed by today's serving stacks.

Serving MLLMs inherits every headache of text-only LLM serving—tight tail-latency [15, 5] targets, massive KV-cache footprints [35, 29, 13], dynamic batching[7, 38], and adds a new one: modality-specific encoders. Most production stacks [13, 41, 10, 1] remain tuned for single-modality workloads. Optimizations such as prefilling decoding disaggregation [11, 42], chunked prefill [1] boost through-

---

*Corresponding author.

39th Conference on Neural Information Processing Systems (NeurIPS 2025).

put for text models assuming a homogeneous stream of decoder kernels. That assumption breaks once a vision or audio encoder, with a completely different resource profile, enters the pipeline.

Lacking encoder-aware machinery, state-of-the-art systems such as vLLM [13] or TGI [10] revert to *time-multiplexing* when they serve MLLMs: the GPU runs the vision/audio encoder first, then switches context and launches the text decoder, and so on. Encoder and decoder kernels therefore contend for the accelerator *sequentially*, losing the complementary compute-versus-memory balance that could otherwise be exploited.

Our key insight is that **encoders and decoders have complementary resource footprints**: the compute cycles a decoder leaves idle are exactly what an encoder can exploit, while the memory an encoder scarcely touches is what the decoder hungers for to house its KV-cache. We therefore advocate **space-multiplexing** instead of traditional time-multiplexing—running encoder and decoder kernels *simultaneously* on the same GPU, each claiming just the compute and memory it truly needs.

We embody this idea in `SpaceServe`, a split-encoder serving that: 1) decouples all modality-specific encoders from the shared text-decoder; 2) dynamically routes multimodal requests through a modality-aware scheduler that adapts to workload mix; 3) achieves space-multiplexing by co-locating encoder and decoder kernels on each GPU whenever their complementary compute-memory footprints align. Our contributions are as follows:

- **Split-encoder serving architecture:** We introduce `SpaceServe`, the first system that *decouples* all modality-specific encoders from the shared text decoder *and* provides a GPU runtime that *co-locates* their kernels via space-multiplexing. A lightweight, cost-model-guided co-location policy packs encoder and decoder kernels whenever their resources footprints fit.
- **TWSRFT encoder scheduler:** To tackle bursty multimodal workloads, we design a *Time-Windowed Shortest-Remaining-First (TWSRFT)* scheduler that orders *encoder* requests within each time window by their remaining work. This preemptive policy minimizes encoder completion latency and smooths the arrival pattern seen by the shared decoder, improving end-to-end tail latency without harming encoder performance.
- **Comprehensive evaluation:** We evaluate `SpaceServe` with 4 size of state-of-the-art MLLMs [36, 3] on three GPU combinations (1*A100, 4*A100, 8*A100) using mixed workloads sampled from two well-known benchmarks for image [40] and video [8]. Empirical results demonstrate that `SpaceServe` cuts *time-per-output-token* (TPOT) by $4.81\mathbf{x}$ on average and up to $\mathbf{28.9x}$.

## 2   Motivation: From *time*-Multiplexing to *space*-Multiplexing

### 2.1   Preliminary

The canonical MLLM stack is organized around two core modules: 1) Modality-specific encoders that translate raw visual, audio, or other non-text inputs into high-dimensional embeddings. 2) A shared text-LLM decoder that performs autoregression generation over these embeddings. A lightweight pre-processing stage precedes the encoders, normalising each modality into the expected tensor format—tokenising text, slicing images into fixed-size patches, or converting audio into waveforms, thus the encoders can operate on a consistent representation. For instance, a vision encoder typically use ViT-based [6] structure to encode visual patches into visual tokens, which are aligned with text tokens in a unified representation space for integration and then processed by the LLM for reasoning services and coherent outputs generation. A classic MLLM architecture is depicted by Figure 1a.

### 2.2   Resource footprint complementary: A quantitative analysis

Our major insight is a clear **resource-footprint complementarity** between modality encoders and the shared text decoder. We substantiate this with a quantitative study of arithmetic intensity (AI)—floating-point operations per transferred byte (FLOPs / Byte)—showing that encoders are compute-rich but memory-light, whereas decoders are memory-hungry yet compute-light. This complementarity underpins the space-multiplexing design that follows.

Due to space limit, the detailed computation of AI for MLLMs are put in Appendix D.

Table 1 reports the arithmetic intensity (AI, FLOPs/byte) of Qwen 2-VL-7B at two image resolutions—$512 \times 512$ and $2048 \times 2048$. Processing such *dynamic* input sizes is crucial for accuracy, yet it complicates serving. Two key take-aways emerge:

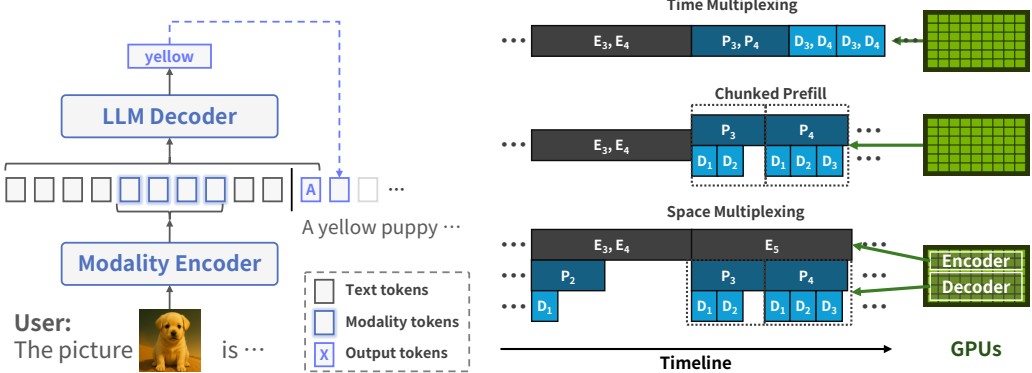

(a) Core components of MLLM: encoder and decoder.

(b) From *time*-Multiplexing to *space*-Multiplexing

Figure 1: Architecture of MLLMs and how vllm and `SpaceServe` serves MLLMs

Table 1: Arithmetic intensity (FLOPs / byte) of each serving stage in Qwen2-VL-7B on A100 GPU as image resolution increases from $512 \times 512$ to $2048 \times 2048$. Higher AI means the kernel is more compute-bound; lower AI implies it is more memory-bound.

| Stage | AI @ $512 \times 512$ | AI @ $2048 \times 2048$ |
|---|---|---|
| Encoder encoding | 338.21 | 8826.97 |
| Decoder prefilling | 72.13 | 1212.43 |
| Decoder decoding | 0.887 | 0.887 |

1. **Encoders and decoders are complementary.** At both resolutions the *encoder encoding* remains compute-bound—AI increases from 338.21 to 8826.97 FLOPs/byte as resolution increases, but even 338.21 still far exceeds the decoder's AI. By contrast, the *decoder decoding* stage stays memory-bound at 0.887 FLOPs/byte. Consequently, an encoder's surplus memory bandwidth can satisfy a decoder's KV-cache needs, while a decoder's idle SM cycles can be reclaimed by encoder kernels—precisely the synergy exploited by our space-multiplexing scheduler.

2. **Input dynamics reshape the resource footprint.** Increasing resolution quadruples the number of $14 \times 14$ vision patches (e.g., $256 \rightarrow 1024$ patches), inflating memory traffic slower than compute and increasing encoder AI by roughly $4\times$, as the AI of encoder is quadratically related to the number of patches, as detailed in Appendix C. The *Encoder* phase changes from $338.21 \rightarrow 8826.97$, and the decode phase is nearly flat, so the *relative* gap between encoder and decoder shrinks as images grow. Any serving stack that statically partitions GPU resources will mis-size one stage or the other once the workload shifts; a scheduler must adapt *per request*, co-locating kernels according to their *current* AI rather than a fixed worst-case estimate.

In short, dynamic, high-resolution inputs amplify the encoder–decoder asymmetry and underscore the need for an adaptive, space-multiplexed serving strategy.

### 2.3 Toward Space-Multiplexed Serving

Existing serving stacks are **modality-agnostic**, so their schedulers treat vision/audio encoders and text decoders as unrelated jobs that simply take turns on the GPU. The result is classic *time-multiplexing* ( Figure 1b, top): the system either runs a batch of encoders or a batch of decoders, never both, squandering the complementary SM-versus-HBM footprints we documented earlier.

State-of-the-art tweaks such as chunked prefill[1] ( Figure 1b, middle) do achieve within-decoder space-multiplexing—interleaving the prefill and decode phases of text requests—but they rely on a critical assumption: both phases operate on the same modality (tokens) with a shared architecture. Vision and audio encoders violate that assumption; they ingest image patches or spectrogram windows and execute a completely different kernel mix, so chunked prefill cannot be repurposed to overlap them with decoder stages.

We therefore argue for true *space-multiplexing across modalities* ( Figure 1b, bottom). Through a GPU-level partitioning runtime and a cost-aware scheduler that allocates sub-GPU slices of SMs (streaming multiprocessors) to each request, our approach co-locates encoders and decoders concurrently, turning their complementary resource profiles into tangible throughput and latency gains.

# 3   `SpaceServe`: **Design**

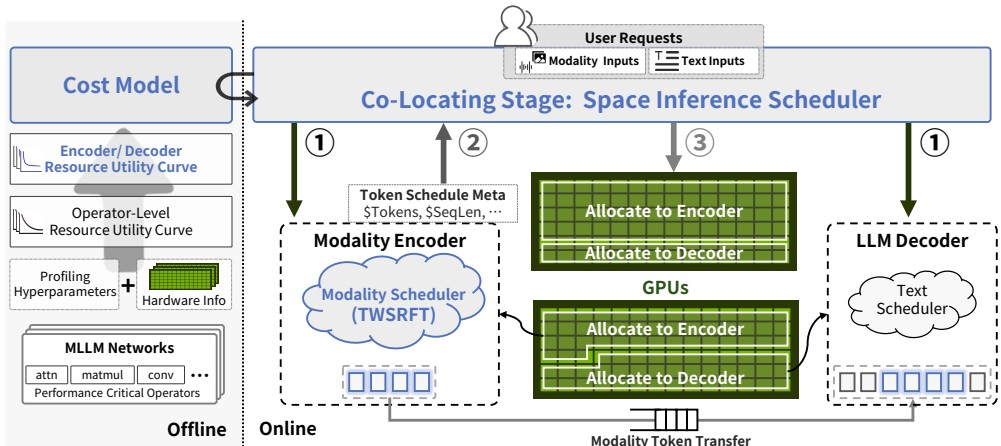

Figure 2: `SpaceServe` System Overview

`SpaceServe` converts the traditional *time*-multiplexed serving pipeline into a *space*-multiplexed one by decoupling modality encoders from the shared text decoder and then *co-locating* their kernels on the same GPUs. By co-locating kernels, hardware resources are optimized, as different kernels demand distinct resource types. For example, compute-intensive kernel utilize more stream multiprocessors (SMs) but require less memory bandwidth, while memory-bound kernels can use the remaining less SMs without losing bandwidth. Strategic sharing of these kernels on the same GPU enhances resource utilization and boosts performance. Figure 2 sketches the high-level architecture of our approach, which decouples encoders from the LLM and co-locate them on the same GPU. `SpaceServe` achieves *space*-multiplexing of GPU resources through two complementary design pillars that operate in a *split–share* fashion: ① **Disaggregated architecture.** `SpaceServe` cleanly decouples modality encoders from the shared text decoder, enabling each component to execute and be scheduled independently.② GPU runtime with a Space-Inference Scheduler (SIS). A novel runtime characterises the resource footprints of encoder and decoder kernels *per request* and co-locates them on the same GPU set. As illustrated in Figure 2, the dedicated encoder pool is fronted by a *time-window* scheduler that minimises encoding latency under fluctuating loads. At the heart of the system, the **SIS** ( §§ 3.3) dynamically partitions streaming multiprocessors (SMs) between encoders and decoders, guided by a cost model ( §§ 3.1) trained offline from extensive profiling of representative encoder and LLM workloads. During runtime, the SIS continuously monitors GPU utilisation; whenever a new request arrives, it rebalances the SM allocation according to the model's recommendation, sustaining high utilisation while respecting tail-latency constraints.

## 3.1   **Hierarchical Operator-level Profiling for Resource-Utility Curves**

Efficient sub–GPU sharing requires a *precise performance model* that can predict how both encoders and decoders behave under *(i)* **dynamic inputs** and *(ii)* **dynamic resource allocations**. Two challenges emerge: 1) **Input-dependent behaviour:** As shown in § 2 and §§ 2.2, changes in image or video resolution alter the arithmetic intensity (AI) of both encoders and decoders, shifting their compute–memory balance. 2) **Non-linear scaling with SM slices.** The latency or throughput of a kernel does *not* scale linearly with the fraction of streaming multiprocessors (SMs) it receives, making naive proportional models inaccurate.

These realities motivate the notion of *Resource–Utility Curves*: explicit mappings from the GPU resources granted to a component (e.g., SM percentage) to the *utility* it delivers (e.g., latency, tokens). Such curves are the foundation of an adaptive serving stack. We construct them via a **hierarchical, operator-level profiling** workflow ( §§ 3.1), which samples the performance of individual operators under controlled SM partitions and then aggregates the results to produce end-to-end utility curves for each encoder and decoder variant. The details of controlling SMs allocation are in §§§ 3.3.2. Thanks to the convergence of operators, we mainly focus on two types of operators:

---

**Algorithm 1** Compose Model-Level Resource-Utility Curve

---

**Input:** ModelComponentType $MCT$
**Input:** ModelArchitecture $Arch$ (details $N_{layers}, d_{model}$, etc.)
**Input:** PrimaryInput $P_{in}$ (a specific $L_{patches}$ or $S_{len}$)
**Input:** SMCountRange $R_{SM}$
**Input:** OpPerfDB
**Output:** ModelCurve[$N_{SM}$] $\rightarrow$ TotalModelLatency for $P_{in}$
 1: ModelCurve $\leftarrow \emptyset$
 2: **for** $N_{SM} \in R_{SM}$ **do**
 3:     TotalModelLatency $\leftarrow 0$
 4:     **for** $i \leftarrow 1$ to $Arch.N_{layers}$ **do**
 5:         latency $\leftarrow 0$ {Latency counter for the $i^{th}$ layer}
 6:         **for** $Op_{layer} \in Arch$.getLayerOps($MCT, i$) **do**
 7:             $D_{layerOp} \leftarrow Arch$.getOpDimensions($Op_{layer}, MCT, P_{in}, i$)
 8:             latency $\leftarrow$ latency + OpPerfDB[$Op_{layer}.Type$][$D_{layerOp}$][$N_{SM}$]
 9:         **end for**
10:         $Layer_{Overheads} \leftarrow Arch$.getLayerOverheads($MCT, i, P_{in}, N_{SM}$)
11:         latency $\leftarrow latency + Lat_{Overheads}$
12:         TotalModelLatency $\leftarrow TotalModelLatency + latency$
13:     **end for**
14:     ModelCurve[$N_{SM}$] $\leftarrow TotalModelLatency$
15: **end for**
16:
17: **return** ModelCurve

---

**Profiling GEMM Resource Utility Curve :** (1)For Encoder: Profile various GEMMs involved in FFN layers (e.g., matrices of size $Lpatches \times d_{enc}$ by $d_{enc} \times 4d_{enc}$) and attention projections (e.g., $Lpatches \times d_{enc}$ by $d_{enc} \times d_{enc}$). For each matrix size, a curve of Latency vs. $Nsm$ is generated. (2)For Decoder: Profile GEMMs for FFN layers (e.g., $1 \times d_{llm}$ by $d_{llm} \times 4d_{llm}$), attention projections for the new token, and the LM head. Each results in a Latency vs. $Nsm$ curve.

**Profiling Attention Resrouce Utility Curve:**(1)For Encoder: Profile the encoder's self-attention mechanism for different $Lpatches$. This also yields a set of Latency vs. $Nsm$ curves, one for each $Lpatches$. (2) For Decoder: Profile the attention mechanism computing a new token's attention against a KV cache of length $Slen$. This generates Latency vs. $Nsm$ curves for various $Slen$.

After the profile, we compose the Model-level resource-utility curves by Algorithm 1.

## 3.2 Encoder Stage: TWSRFT Scheduler

In line with dynamic batching [38] for text only decoders, `SpaceServe` also incorporate a modality-aware scheduler for batching encoder requests. The choice of requests to batch significantly affects latency metrics, such as Time to First Token (TTFT) and Time per Output Token (TPOT). For instance, encoding a high-resolution image typically takes longer than processing lower-resolution image embeddings. To minimize latency and boost throughput, batching smaller requests is effective. However, this approach faces two constraints. First, GPU computational capacity limits batching, as encoders are compute-intensive. For example, the NVIDIA A100 has an optimal arithmetic intensity of approximately 161 FLOPs/Byte. The threshold varies across GPU models. Second, prioritizing small requests can starve larger requests, delaying their processing.

To address these challenges, `SpaceServe` introduces the Time-Windowed Shortest Remaining Time First (TWSRTF) scheduler, the modality scheduler shown in Figure 2. The TWSRTF scheduler uses

a time-window approach to manage request batching. Requests are collected in an input queue within a fixed time window. Within this window, requests are batched by size, prioritizing the smallest ones first. The scheduler also monitors batch capacity to ensure its stay within the GPU's limits. The detailed algorithm implementation is presented in Algorithm 2

---

**Algorithm 2** Time-Windowed Shortest Remaining Time First Scheduling for Encoder

---

**Input:** Request queue $Q$, window size $w$, maximum patches $seqlength_{batchsuitable}$
**Output:** Encoded batches of requests
 1: **while** $Q$ is not empty **do**
 2:    $W \leftarrow Q$.get($w$) {Fetch a window of requests}
 3:    Sort $W$ by ascending patches
 4:    $i \leftarrow 0; B \leftarrow [\,W[i]\,]; s \leftarrow W[i]$.patches
 5:    $i \leftarrow i + 1$
 6:    **while** $i < |W|$ **do**
 7:      **if** $s + W[i]$.patches $< seqlength_{batchsuitable}$ **then**
 8:        Append $W[i]$ to $B$
 9:        $s \leftarrow s + W[i]$.patches
10:        $i \leftarrow i + 1$
11:      **else**
12:        encoder($B$) {Process batch with encoder}
13:        Remove $B$ from $Q$
14:      **end if**
15:    **end while**
16: **end while**

---

### 3.3 Co-Locating Stage: Space Inference Scheduler for Space Multiplexing

The Online Space Inference Scheduler is a cornerstone of the Co-Locating Stage, dynamically managing request execution flows and the partitions of Streaming Multiprocessors (SMs) between encoders and the LLM decoders based on runtime conditions. It oversees the scheduling of incoming requests and ensures seamless coordination with the Modality Encoder Scheduler. When the Modality Encoder Scheduler, using Time-Windowed Shortest Remaining Time First (TWSRTF), dispatches a new encoding request, it triggers an action in the Online Space-Multiplex Scheduler. The scheduler utilizes the request's metadata, e.g. input sequence length and image resolution, alongside the cost model derived from offline profiling (detailed in §§ 3.1). This enables dynamic allocation of GPU resources of the same processor, particularly SMs, between the encoder task and current LLM decoding operations. Such adaptive, request-aware resource allocation optimizes performance and maximizes hardware utilization for diverse multimodal workloads.

#### 3.3.1 GPU Runtime: Fine-grained Resource Partition

Effective space-multiplexing hinges on the ability to partition a single GPU and allocate resources *below* the device boundary. Modern accelerators now expose precisely this functionality. The AMD HIP runtime, for example, supports CU masks [2] that assign disjoint CU sets to different streams, while NVIDIA GPUs offer comparable streaming-multiprocessor control via `libsmctrl` [4]. `SpaceServe` builds on these sub-GPU partitioning primitives to realise fine-grained resource allocation.

#### 3.3.2 Request-aware SM Partition

During the serving phase of the inference engine, we design an online allocation mechanism based on our profiler to dynamically optimize Streaming Multiprocessor (SM) partitioning.

`SpaceServe` decomposes the inference service of multimodal large models into three independent processes: a CPU process, an Encoder process, and a LLM process. The CPU process handles user requests and performs preprocessing of multimodal data. Request information is then dispatched separately to both the Encoder process and the LLM process. Given that the Encoder and LLM have distinct resource preferences and requirements, `SpaceServe` applies different scheduling strategies tailored to each. `SpaceServe` performs dynamic and efficient GPU resource partitioning tailored to incoming requests. This adaptive allocation is underpinned by our comprehensive Model Resource

Utility Curves in §§ 3.1. The underlying Streaming Multiprocessor (SM) partitioning algorithm, which is elaborated in Appendix E operates on the core principle of finding the resource configuration that minimizes the sum of the Encoder's and Decoder's execution times.

As for the LLM Decoder, `SpaceServe` employs a ChunkedPrefill [1] with vision cache scheduling strategy for the LLM processing, which is formally described in Algorithm 3.

# 4 Evaluation

## 4.1 Experimental Setup

**Hardware**: We evaluate `SpaceServe` on a server which is equipped with 8 NVIDIA A100 SXM GPUS and Intel Xeon(R) Gold 6430 CPU.

**Models and workloads**: We select 4 state-of-the-art MLLMs with varying model size, including Qwen2-VL-2B [36], Qwen2-VL-7B [36], Qwen2.5-VL-32B [3] and Qwen2-VL-72B [36]. These models were chosen for their proficiency in handling high-resolution images with arbitrary aspect ratios, understanding long contextual sequences. Due to GPU-memory constraints, we run Qwen2-VL-2B and Qwen2-VL-7B on a single GPU, Qwen2.5-VL-32B on four GPUs, and the largest Qwen2-VL-72B on eight GPUs. For the multi-GPU configurations we employ *tensor parallelism*.

**Datasets**: We build a 1,740-example evaluation set by sampling from MMMU-Pro [40] and Video-MME [8], preserving an 8 : 2 ratio of image to video items.

**Baseline Systems**: We adopt `vLLM` [13]—a SOTA serving framework—as our baseline. In particular, we evaluate against the latest `vLLM v1` architecture, which integrates recent advances such as *zero-overhead scheduling* and a suite of MLLM-specific optimizations.

**Metrics**: LLM inference efficiency centers on two critical latency metrics: Time To First Token (TTFT) and Time Per Output Token (TPOT), which we use as primary metrics. Lower values for both TTFT and TPOT indicate better performance.

## 4.2 End-to-End Evaluation Results

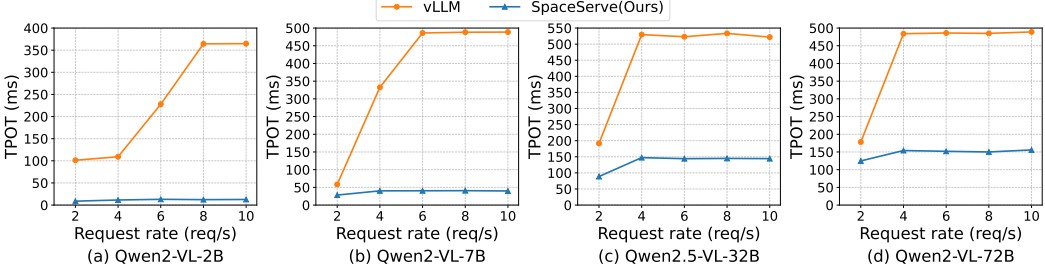

Figure 3: TPOT of `SpaceServe` and vLLM on various models, lower TPOT↓ is better

In the experiment, we increased the request rate from 2 to 10 requests per second. Across various model sizes, ranging from 7B to 72B, `SpaceServe` consistently and significantly outperforms the vLLM baseline. For instance, while serving the Qwen2-VL-2B model, vLLM's average TPOT increased from 101ms at 2 RPS to 365ms at 10 RPS, indicating a significant drop in efficiency. In contrast, `SpaceServe` demonstrated robust performance, with TPOT increasing only slightly from 8.85ms (2 RPS) to 12.62ms (10 RPS). This represented a substantial improvement, achieving a 28.9x reduction in TPOT compared to vLLM at 10 RPS and thereby enhancing service throughput and user experience.

This trend continued across other models at 10 RPS. For Qwen2-VL-7B, `SpaceServe` recorded a TPOT of 40ms compared to vLLM v1's 489ms, delivering a speedup of 12.3x. For larger models with greater resource demands, such as Qwen2.5-VL-32B on four NVIDIA A100 GPUs, `SpaceServe`'s TPOT was 144ms compared to vLLM's 522ms (10 RPS), a speedup of 3.62x. Similarly, when serving Qwen2-VL-72B on eight NVIDIA A100 GPUs at 10 RPS, `SpaceServe` maintained its advantage with a TPOT of 155.8ms versus vLLM v1's 489ms, achieving a 3.14x speedup.

**Why do we have such an excellent token generation speed:** The inherent latency for generating a single token during the decoder phase is typically low, around 10 milliseconds for Qwen2VL-2B. However, vLLM-v1's time-division multiplexing architecture causes decoding requests to be blocked when the GPU is occupied by an encoder task, which can take several hundred milliseconds, e.g. 671ms for Qwen2VL-2B. This forces decoding operations into a wait state until the encoder task completes, significantly inflating the Time Per Output Token (TPOT) in high request-per-second (RPS) scenarios due to frequent encoder invocations.

In stark contrast, `SpaceServe` employs a spatial multiplexing strategy (or a functionally equivalent mechanism that allows for concurrent or dedicated resource allocation for encoder and decoder stages). This architectural design effectively decouples the high-throughput decoder operations from the more time-intensive encoder tasks. By mitigating these inter-dependencies, `SpaceServe` ensures that the token generation rate for the decoder remains stable and consistently high, achieving approximately 100 tokens per second, even during frequent encoder activity. This resilience of the decoding pipeline to encoder-induced stalls is the primary contributor to `SpaceServe`'s superior performance under demanding, mixed encoder-decoder workloads when measured in TPOT.

Due to page constraints, we put the TTFT results on Appendix B. Unlike the sharp TPOT gains, the time-to-first-token (TTFT) remains virtually unchanged between `SpaceServe` and vLLM for all four models. This is expected: `SpaceServe`'s design focuses on harvesting the memory-bandwidth slack present while an encoder is running to accelerate the memory-bound decoder, boosting steady-state throughput rather than the initial-token latency captured by TTFT.

### 4.3 Ablation Study

The performance advantages of `SpaceServe` compared to vLLM are attributed to two primary innovations: *(1) the efficient space-multiplexing methodology. (2) the specialized Time-Windowed Shortest Remaining Time First (TW-SRTF) scheduler for the vision encoder.* To delineate the individual contributions of these elements, we performed targeted ablation studies, adhering to the experimental setup described earlier.

#### 4.3.1 Ablation Study: Space Inference Scheduler vs. MPS

We ablate our space-multiplexing design against NVIDIA Multi-Process Service (MPS) [20], using Qwen2-VL-7B as the test model.

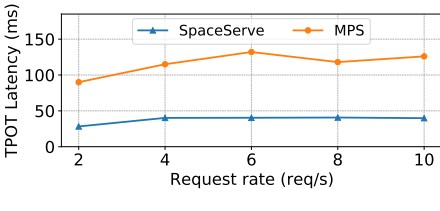
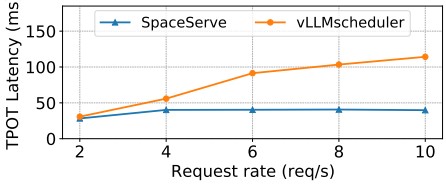

(a) TPOT compared with MPS      (b) TPOT compared with vLLM scheduler

Figure 4: Ablation study of Qwen-2-VL-7B with TPOT

As shown in Figure 4, the ablation study confirms our hypothesis: Compared to vLLM, the MPS (Multi-Process Service) version of `SpaceServe` accelerates the TPOT metric while keeping a largely unaffected TTFT. Our ablation study further evaluates our `SpaceServe` model, which employs a fine-grained allocation strategy, against this MPS version. At an input rate of 2 Requests Per Second (RPS), the MPS version of `SpaceServe` shows a TPOT of 90 ms. This latency increases with rising RPS, reaching a peak of 132 ms. In contrast, our `SpaceServe` (with the fine-grained strategy) achieves a TPOT of 28.25 ms at 2 RPS, and its TPOT only reaches a maximum of 40.68 ms as RPS increases. This comparison highlights that our fine-grained allocation strategy maintains significantly lower TPOT latency and greater stability under high concurrent requests, achieving a TPOT speedup of up to 3.3x relative to the MPS version. Moreover, as depicted in Figure 5a and Figure 5b, the TTFT for both approaches is nearly identical, indicating that these TPOT enhancements do not compromise first-token latency.

The substantial performance improvement stems from alleviating resource contention at the SM microarchitectural level. When Encoder and LLM processes are co-located using only NVIDIA MPS, their respective CUDA kernels, comprised of multiple warps, may be concurrently scheduled by the MPS server onto any available SM. Within a single SM, these distinct workloads then vie for limited resources such as arithmetic units, register file space, L1 cache, and shared memory bandwidth. The disparate nature of these tasks—the Encoder being typically compute-bound and the LLM decoding phase often exhibiting bursty memory access patterns and varied computational demands—can lead to suboptimal SM utilization. For example, high register usage by one task could limit the number of active warps (occupancy) for the other, or differing memory access patterns could lead to L1 cache pollution, increasing effective memory latency.

By contrast, `SpaceServe`'s SM partitioning dedicates distinct sets of SMs to the Encoder and LLM processes. This spatial isolation minimizes direct inter-process contention for intra-SM resources. Each process can therefore more effectively exploit the full capacity of its allocated SMs, leading to improved instruction issue rates and better sustained occupancy for its specific workload characteristics. This dedicated execution environment allows for more streamlined processing within each SM partition, reducing stalls and ultimately translating to a lower TPOT and an enhanced throughput.

### 4.3.2   Ablation Study: TWSRFT Scheduler vs. vLLM Default Scheduler

We ablate our redesigned encoder scheduler—built for split-encoder architectures—using the Qwen2-VL-7B model.

The results demonstrate a clear advantage for our custom scheduling approach under increasing load. At a low concurrency of 2 Requests Per Second (RPS), the `SpaceServe-vLLMscheduler` exhibited a TPOT of 30.73 ms, which was only slightly higher than our full `SpaceServe` system's TPOT of 28.25 ms. However, as the concurrency increased to 4 RPS, the TPOT for `SpaceServe-vLLMscheduler` rose more sharply to 55.88 ms, whereas our `SpaceServe` system maintained a TPOT of 40.18 ms. This performance disparity became significantly more pronounced at 10 RPS: the TPOT for `SpaceServe-vLLMscheduler` surged to 114.22 ms, while our `SpaceServe` system impressively sustained a low TPOT latency of 39.8 ms, showcasing its stability and efficiency at higher throughputs. The observed degradation in TPOT for the `SpaceServe-vLLMscheduler` configuration can be attributed to the vLLM scheduler's lack of an independent batching strategy for the encoder component. When the aggregated batch size of vision tokens processed by the encoder reaches a critical threshold, any further increase in the number of concurrent requests leads to a substantial rise in the overall execution time for the encoder stage. This, in turn, inflates the average processing time per request. Consequently, the LLM process does not receive a sufficient or timely stream of vision tokens, which acts as a bottleneck and significantly slows down the generation of subsequent output tokens, thereby increasing the TPOT.

### 4.4   Performance on Modern MoE-based Architectures

To demonstrate `SpaceServe`'s effectiveness on the latest generation of MLLMs, we evaluated its performance on cutting-edge models featuring sparse MoE layers: **DeepSeek-VL2** and **Kimi-VL**. The experiments were run under a high-concurrency load of 10 requests per second.

The results, presented in Table 2, show that `SpaceServe` delivers substantial performance gains. We achieve a **4.08x** TPOT speedup on DeepSeek-VL2 and a remarkable **9.84x** speedup on Kimi-VL.

This dramatic performance improvement, especially on Kimi-VL, is attributed to its exceptionally compute-intensive vision encoder, which creates a significant imbalance with its memory-bound MoE decoder. This is precisely the scenario where SpaceServe's spatial multiplexing provides the greatest benefit, by co-locating the complementary kernels to maximize GPU utilization.

### 4.5   Performance Scaling with Input Resolution

Modern MLLMs increasingly use high-resolution inputs to improve understanding. This makes the vision encoder substantially more compute-intensive, creating a major bottleneck for traditional serving systems. We evaluated SpaceServe from 224x224 to 2Kx2K under high concurrency (10 RPS). As shown in Table 3, vLLM's TPOT rises sharply with resolution because the long-running encoder blocks the decoder, whereas SpaceServe's TPOT stays low and stable.

Table 2: TPOT speedup on MoE-based MLLMs under a high-concurrency workload (10 RPS).

| Model (MoE-based) | Framework | TTFT (ms) | TPOT (ms) | TPOT Speedup |
|---|---|---|---|---|
| DeepSeek-VL2 | vLLM-v1 | 5597 | 122.5 | Baseline |
| | **SpaceServe (Ours)** | 4495 | 30.0 | **4.08x** |
| Kimi-VL | vLLM-v1 | 57900 | 482.0 | Baseline |
| | **SpaceServe (Ours)** | 56323 | 49.0 | **9.84x** |

Crucially, the performance advantage of SpaceServe scales with the computational load. As illustrated in Table 3, the TPOT speedup escalates from **1.37x** on low-resolution inputs to an impressive **12.39x** at 2K resolution. This demonstrates that spatial multiplexing is most effective precisely where it is most needed: mitigating the latency of compute-heavy tasks. Far from being a limitation, high-resolution inputs highlight SpaceServe's fundamental strength in harnessing resource heterogeneity to deliver efficient performance for the next generation of high-fidelity MLLMs.

Table 3: Performance comparison across varying input resolutions at 10 RPS. As resolution increases, SpaceServe maintains a stable, low TPOT, while the baseline's latency degrades significantly.

| Resolution | Framework | TTFT (s) | TPOT (ms) | TPOT Speedup |
|---|---|---|---|---|
| $224 \times 224$ | vLLM-v1 | 0.09 | 23.6 | Baseline |
| | **SpaceServe (Ours)** | 0.09 | 17.2 | **1.37x** |
| $512 \times 512$ | vLLM-v1 | 0.21 | 38.6 | Baseline |
| | **SpaceServe (Ours)** | 0.21 | 25.9 | **1.49x** |
| $1K \times 1K$ | vLLM-v1 | 15.31 | 212.6 | Baseline |
| | **SpaceServe (Ours)** | 15.54 | 35.6 | **5.97x** |
| $2K \times 2K$ | vLLM-v1 | 139.35 | 470.7 | Baseline |
| | **SpaceServe (Ours)** | 135.76 | 38.0 | **12.39x** |

## 5 Related Work

**Disaggregated Serving.** Promising for large models, disaggregated serving (e.g., SplitWise [22], DistServe [42], DéjàVu [31]) decouples prefill/decode stages, mitigating interference for better TTFT/TPOT control. These LLM-focused systems often neglect the MLLM-specific encoding step. Even recent extensions (Pensieve [39], Mooncake [23], PD-Serve [11]) with advanced KV cache management offer limited MLLM applicability.

**Multi-modality Model Serving.** Serving multimodal models (MLLMs) typically relies on adapting LLM systems (e.g., vLLM [13], SGLang [41]) or using recent MLLM inference code [24]. Adapting LLM systems hits LMM encoding bottlenecks with rich multimedia. Techniques like KV cache eviction [14, 19] and compression [12] are often model/scenario-specific (e.g., Inf-MLLM's [19] single-GPU streaming) and may miss cloud SLOs. Early disaggregation ideas (e.g., EPD [30]) lack `SpaceServe`'s comprehensive scheduling and partitioning.

**Complementarity with Sparse MLLM Architectures.** `SpaceServe` complements sparse multimodal architectures, including Mixture-of-Experts designs [33, 37, 16]. Sparsity can lower per-request compute, but it also increases resource heterogeneity by leaving hardware assigned to inactive experts underutilized. `SpaceServe` exploits this opportunity by co-locating the compute-intensive encoder of one request with the sparse, memory-bound decoder of another, using spatial multiplexing to fill idle resources and raise overall utilization

## 6 Conclusion

This paper introduces `SpaceServe`, a novel MLLM serving system featuring two key innovations: fine-grained GPU resource management and an advanced scheduling framework. This design enables concurrent low-latency inference and high-throughput for multimodal LLMs, outperforming SOTA systems by up to 28.9 ×.

## Acknowledgments

This work was supported by the National Key R&D Program of China (Grant No. 2024YFB4505603) and by the National Natural Science Foundation of China (Grant Nos. U23B2020, 62090024, and 62302479).

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

## A  Limitations and future work

While `SpaceServe` effectively addresses the challenge of extended Time Per Output Token (TPOT) in multimodal large language model inference, another critical performance indicator, Time To First Token (TTFT), remains a significant consideration. TTFT directly reflects the system's initial responsiveness and is paramount for a positive user experience. In its current state, `SpaceServe` maintains TTFT performance comparable to existing state-of-the-art frameworks, such as vLLM-v1. However, substantially reducing the execution latency of a single encoder pass—the primary determinant of TTFT—continues to be a demanding task. Optimizing this single-encoder latency to further enhance TTFT represents an important avenue for our future research.

## B  Experiments for TTFT metric

Table 4: Time To First Token (TTFT) comparison across different models with varying Request Per Second (RPS), lower TTFT↓ is better

| Method | Time To First Token (TTFT) (s) ↓ | | | | |
|---|---|---|---|---|---|
| RPS | 2 | 4 | 6 | 8 | 10 |
| **Qwen-2-VL-2B** | | | | | |
| vLLM | 12.068 | 16.180 | 18.655 | 23.035 | 26.038 |
| `SpaceServe` | 13.324 | 17.515 | 21.114 | 24.726 | 27.437 |
| **Qwen-2-VL-7B** | | | | | |
| vLLM | 19.492 | 22.033 | 30.306 | 34.745 | 38.795 |
| `SpaceServe` | 18.832 | 22.419 | 32.050 | 36.533 | 39.086 |
| **Qwen-2.5-VL-32B** | | | | | |
| vLLM | 16.553 | 32.631 | 41.849 | 46.440 | 47.547 |
| `SpaceServe` | 17.131 | 33.242 | 42.489 | 47.144 | 48.966 |
| **Qwen-2-VL-72B** | | | | | |
| vLLM | 14.872 | 26.647 | 35.717 | 39.780 | 42.067 |
| `SpaceServe` | 14.810 | 26.633 | 36.321 | 38.378 | 43.834 |

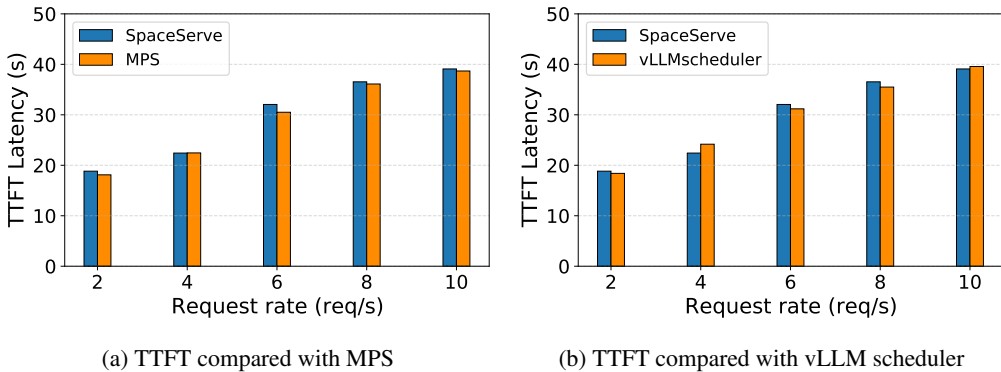

(a) TTFT compared with MPS      (b) TTFT compared with vLLM scheduler

Figure 5: Ablation study of Qwen-2-VL-7B with TTFT

## C  Arithmetic Intensity Modeling for Encoder

Taking vision encoders for example, the patch generation characteristics can be formulated as:

$$H' = S \left\lceil \frac{H}{S} \right\rceil, \quad W' = S \left\lceil \frac{W}{S} \right\rceil \quad \Rightarrow \quad L_{\text{patches}} = \left\lceil \frac{H'}{L} \right\rceil \times \left\lceil \frac{W'}{L} \right\rceil \tag{1}$$

where S denotes image padding strides, H denotes the original image height, H' denotes the normalized height, W denotes the original image width, and W' denotes the normalized width. And L denotes the patch size, $L_{\text{patches}}$ denotes the total patches after preprocessing. During the encoder stage, the FLOPs can be formulated as :

$$\text{FLOPs}_{\text{vision}} = (\underbrace{2L_{\text{patches}}d_{\text{vision}}}_{\text{Embedding}} + \underbrace{8L_{\text{patches}}d_{\text{vision}}^2 + 4L_{\text{patches}}^2 d_{\text{vision}}}_{\text{Attention}} + \underbrace{16L_{\text{patches}}d_{\text{vision}}^2}_{\text{FFN}}) * Layers \quad (2)$$

The overall Arithmetic Intensity can be approximated by:

$$\frac{\text{FLOPs}_{\text{vision}}}{\text{Memory}_{\text{weight}}} \quad (3)$$

## D    Arithmetic Intensity Modeling for LLM

**LLM Prefill Stage**    Before autoregressive token generation, the Large Language Model (LLM) processes the entire input prompt (with sequence length $S_{prompt}$) in a single, parallel forward pass. This 'prefill' stage is essential for computing the initial set of Key-Value (KV) states for all tokens in the prompt. To optimize this process, particularly for managing the memory footprint and computational cost of the attention mechanism, techniques like Grouped-Query Attention (GQA) are often employed. These initial KV states, potentially generated using GQA, are then cached and utilized by the subsequent decoding steps.

**LLM Prefill: FLOPs.**  Let $L_{llm}$ be the number of LLM layers, $d_{llm}$ be the hidden dimension (consistent with the decoding stage notation), $S_{prompt}$ be the input prompt length, and $V$ be the vocabulary size. The FFN expansion factor is $c$ (typically $c = 4$). The FLOPs for the Transformer backbone during prefill are primarily from self-attention and FFN computations:

- *Self-Attention per layer:* $\approx (8S_{prompt}6d_{llm}^2 + 4S_{prompt}^2 d_{llm})$ FLOPs. This encompasses Q,K,V projections ($6S_{prompt}d_{llm}^2$), attention score calculations ($QK^T$) and application to Values ($4S_{prompt}^2 d_{llm}$), and the output projection ($2S_{prompt}d_{llm}^2$).
- *FFN per layer:* $\approx 4cS_{prompt}d_{llm}^2$ FLOPs. For $c = 4$, this is $16S_{prompt}d_{llm}^2$.

The total FLOPs for the Transformer blocks during prefill ($FLOPs_{Prefill\_Transformer}$) are:

$$FLOPs_{Prefill\_Transformer} \approx L_{llm}((8 + 4c)S_{prompt}d_{llm}^2 + 4S_{prompt}^2 d_{llm}) \quad (4)$$

Assuming $c = 4$:

$$FLOPs_{Prefill\_Transformer} \approx L_{llm}(24S_{prompt}d_{llm}^2 + 4S_{prompt}^2 d_{llm}) \quad (5)$$

If output logits are computed for the entire prompt (e.g., for training or certain inference strategies), the LM head adds $FLOPs_{LM\_Head\_Prefill} \approx 2S_{prompt}d_{llm}V$.

**LLM Prefill: Memory Access.** Memory operations during prefill include:

- *Model Weights ($Memory_{weights\_LLM}$):*   Accessing the model parameters $N_{total\_params\_LLM}$, where $Memory_{weights\_LLM} \approx N_{total\_params\_LLM} \times$ bytes_per_parameter.
- *Input Data & Activations:* Reading input prompt embeddings and handling intermediate activations for the $S_{prompt}$ length sequence across layers.
- *KV Cache Generation:* Writing the KV pairs for all $S_{prompt}$ tokens. The size of this cache is $Memory_{KV\_cache\_prompt} = S_{prompt} \times L_{llm} \times 2 \times d_{llm} \times$ bytes_per_element.

**LLM Prefill: Arithmetic Intensity (AI).** The Arithmetic Intensity for prefill is $AI_{Prefill} =$ Equation 5/(Memory Access for Weights, Input, Activations, and KV Cache Write).

$$AI_{Prefill} = \frac{L_{llm}((8 + 4c)S_{prompt}d_{llm}^2 + 4S_{prompt}^2 d_{llm})}{Memory_{Access\_Prefill}} \quad (6)$$

The $4L_{llm}S^2_{prompt}d_{llm}$ term in the FLOPs (Equation 5) means that for long prompts ($S_{prompt}$), the prefill AI can be substantial, potentially making this phase compute-intensive. This contrasts with the characteristics of the subsequent decoding phase.

In contrast, the Arithmetic Intensity of the LLM component is significantly lower. The LLM decoding stage operates autoregressively, generating one token at a time. Its computational characteristics are modeled as follows:

**LLM Decoding: FLOPs (Per Token Generation)**   For the generation of a single token, let $L_{llm}$ be the number of LLM layers, $d_{llm}$ be the hidden dimension, $S_{len}$ be the current sequence length (context from previously generated tokens), and $V$ be the vocabulary size. The primary FLOPs contributions are:

- **Transformer Decoder Layers (with KV Cache):** For each token, each layer involves:
    - Self-Attention (new token Q, K, V projections; attention with cached K, V): Approximately $8d^2_{llm} + 4S_{len}d_{llm}$ FLOPs. This includes Q, K, V projections for the current token ($\approx 6d^2_{llm}$), interaction with the $S_{len}$ cached tokens ($\approx 4S_{len}d_{llm}$ for $QK^\top$ and score-V multiplication), and the output projection ($\approx 22d^2_{llm}$).
    - Feed-Forward Network (FFN): Typically $16d^2_{llm}$ FLOPs (assuming a $4 \times d_{llm}$ intermediate expansion).

    Thus, per layer FLOPs are approximately $24d^2_{llm} + 4S_{len}d_{llm}$.
- **LM Head:** Mapping the final hidden state to vocabulary logits: $\approx 2d_{llm}V$ FLOPs.

The total FLOPs for generating a single token ($FLOPs_{LLM\_decode\_token}$) can be expressed as:

$$FLOPs_{LLM\_decode\_token} \approx L_{llm}(24d^2_{llm} + 4S_{len}d_{llm}) + 2d_{llm}V \tag{7}$$

A common simplified approximation for FLOPs per token (especially when $S_{len}$ is moderate) is $2 \times N_{params\_non\_embedding}$, where $N_{params\_non\_embedding}$ are the non-embedding parameters of the LLM, primarily from FFN and attention projection matrices.

**LLM Decoding: Memory Access (Per Token Generation)**   Memory access during LLM decoding is dominated by:

- **Model Weights ($Memory_{weights\_LLM}$):** Parameters of the Transformer layers and LM head.

$$Memory_{weights\_LLM} \approx N_{total\_params\_LLM} \times \text{bytes\_per\_parameter} \tag{8}$$

- **KV Cache ($Memory_{KV\_cache}$):** Stores Key (K) and Value (V) vectors for $S_{len}$ previous tokens across $L_{llm}$ layers.

$$Memory_{KV\_cache} = S_{len} \times L_{llm} \times 2 \times d_{llm} \times \text{bytes\_per\_element} \tag{9}$$

The critical dynamic memory access per token involves reading from and writing to this KV cache, along with accessing the relevant model weights for the current token's computation.

**LLM Decoding: Arithmetic Intensity (AI)**   The Arithmetic Intensity for LLM decoding is $FLOPs_{LLM\_decode\_token}/Memory_{Access\_per\_token}$. If we primarily consider the model weights for memory access (as their total size is loaded and parts are accessed for each token generation), a simplified AI is:

$$AI_{LLM\_decode} \approx \frac{2 \times N_{params\_non\_embedding}}{N_{params\_total\_LLM} \times \text{bytes\_per\_parameter}} \tag{10}$$

Given that $N_{params\_non\_embedding}$ is a large fraction of $N_{params\_total\_LLM}$, and FLOPs for core matrix multiplies are roughly $2\times$ parameters involved, if we consider all parameters:

$$AI_{LLM\_decode} \approx \frac{2 \times N_{params\_total\_LLM}}{N_{params\_total\_LLM} \times \text{bytes\_per\_parameter}} = \frac{2}{\text{bytes\_per\_parameter}} \tag{11}$$

For FP16/BF16 (2 bytes/parameter), $AI_{LLM\_decode} \approx 1$ FLOP/Byte. This low AI signifies that LLM decoding is predominantly **memory-bandwidth bound**. The substantial memory footprint of weights and the dynamically accessed KV cache, relative to the per-token computational work, contributess to this characteristic.

# E Chunked Prefill Scheduling with Vision Cache Check

**Pre-Scheduling Check**:  Prior to each scheduling iteration, the scheduler examines the $Queue_{encoder\_result}$ to determine if any requests have completed encoding. Completed encoding results from the Encoder process are cached in the LLM process's vision token cache ChunkedPrefill Execution:

**During scheduling of prefill-stage requests**: The scheduler verifies whether the candidate tokens contain vision tokens If required vision tokens are not present in the vision cache: Only the preceding text prompt portion is scheduled Vision token processing is deferred until encoding completion

**Scheduling Priority:** Strict FCFS ordering maintains request sequence integrity Within the Chun-kedPrefill framework: Decoding-stage requests receive scheduling priority over prefill requests. This ensures optimal throughput while preserving fairness

---

**Algorithm 3** Chunked Prefill Scheduling with Vision Cache Check

---

**Input:**
- Waiting prefill requests queue $Q_{prefill}$
- During decoding requests queue $Q_{decoding}$
- Token budget $B$ (max_num_batched_tokens)
- Chunk prefill size $Batch_{chunksize}$
- Vision cache $\mathcal{V}$
- Chunking enabled flag $C$

**Output:** Scheduled batch $Batch$ containing decode and prefill chunks
1: Initialize empty batch: $Batch \leftarrow []$
2: **for** each request $d$ in $Q_{decoding}$ **do**
3:     Append $d$ to $Batch$
4:     Update $B \leftarrow B-$ tokens needed by $d$
5: **end for**
6: **while** $B > 0$ and $Q_{prefill}$ not empty **do**
7:     $r \leftarrow Q_{prefill}$.peek() {Next prefill request}
8:     $v_{pos} \leftarrow$ position of first vision token in $r$
9:     **if** $v_{pos}$ exists and vision tokens at $v_{pos}$ not in $\mathcal{V}$ **then**
10:         $t \leftarrow$ number of prompt tokens before $v_{pos}$
11:         Schedule prefill chunk with tokens $[start_{pos}, v_{pos})$ in $Batch$
12:     **else**
13:         $t \leftarrow$ tokens needed for next chunk of $r$
14:         **if** $t > B$ and $C$ is enabled **then**
15:             Split $r$ into chunk of size exactly $B$ (take next $B$ tokens)
16:             Schedule this chunk for prefill in $Batch$
17:             Remove these $B$ tokens from $r$
18:             $t \leftarrow B$
19:             **break**
20:         **else if** $t \leq B$ **then**
21:             Schedule full prefill request $r$ in $Batch$
22:             Remove $r$ from $Q_{prefill}$
23:         **else**
24:             **break** {No budget left}
25:         **end if**
26:     **end if**
27:     $B \leftarrow B - t$
28: **end while**
29: Execute scheduled $Batch$ (decode + chunked prefill)

---

# F  SM Partition Algorithm

---

**Algorithm 4** FindOptimalSMPartition

---

**Input:**
1: $M_E$ {Encoder request metadata} $RC_E$ {Encoder Resource-Utility Curve}
2: $L_{ctx}$ {Decoder context length} $C_D$ {Decoder chunk configuration} $RC_D$ {Decoder Resource-Utility Curve}
3: $SM_{total}$ {Total allocatable SMs} $SM_{min\_alloc}$ {Minimum SMs per component (e.g., $\geq 1$)}
**Output:** $(sm_E^*, sm_D^*)$ {Optimal SM allocation for Encoder, Decoder}
4: Let $Lat_E(s_E)$ be the latency function for Encoder derived from $RC_E$ using $M_E$.
5: Let $Lat_D(s_D)$ be the latency function for Decoder derived from $RC_D$ using $C_D, L_{ctx}$.
6: $sm_E^* \leftarrow$ null
7: $sm_D^* \leftarrow$ null
8: $L_{makespan}^* \leftarrow \infty$
9: $found\_valid\_allocation \leftarrow$ false
10: **for** $s_E$ from $SM_{min\_alloc}$ to $SM_{total} - SM_{min\_alloc}$ **do**
11:     $s_D \leftarrow SM_{total} - s_E$
12:     **if** $s_D < SM_{min\_alloc}$ **then**
13:         **continue**
14:     **end if**
15:     $current\_L_{makespan} \leftarrow \max(Lat_E(s_E), Lat_D(s_D))$
16:     **if** $current\_L_{makespan} < L_{makespan}^*$ **then**
17:         $L_{makespan}^* \leftarrow current\_L_{makespan}$
18:         $sm_E^* \leftarrow s_E$
19:         $sm_D^* \leftarrow s_D$
20:         $found\_valid\_allocation \leftarrow$ true
21:     **end if**
22: **end for**
23: **if** $found\_valid\_allocation$ **then**
24:     **return** $(sm_E^*, sm_D^*), L_{makespan}^*$
25: **else**
26:     **return** $(\text{null}, \text{null}), \infty$ {No valid allocation found}
27: **end if**

---

# G  Implementation Details

**System stack and versions.**  `SpaceServe` is implemented on top of **vLLM 0.7.2**. We integrate at the scheduler and engine layers, keeping the model execution interface unchanged so that existing MLLM checkpoints can be served without modification.[2]

**Concurrency via CUDA MPS.**  We enable **NVIDIA CUDA Multi-Process Service (MPS)** so that the vision encoder and the text decoder run as separate worker processes on the same GPU and submit kernels concurrently..

**SM partitioning primitive.**  To realize space multiplexing, SpaceServe uses `libsmctrl` as the low-level primitive to partition GPU Streaming Multiprocessors (SMs) across worker processes. The Space-Inference Scheduler (SIS) chooses an SM quota for each worker according to a resource-utility curve and the current TWSRFT admission state, then applies the quota through `libsmctrl`.

---

[2]Code: `https://github.com/gofreelee/SpaceServe`

