# OpenReview forum: "SpaceServe: Spatial Multiplexing of Complementary Encoders and Decoders for Multimodal LLMs"
_NeurIPS.cc/2025/Conference — NeurIPS 2025 poster_

### Official Review · Reviewer_aTid · 2025-07-02

**Clarity:** 4
**Significance:** 4
**Originality:** 4
**Rating:** 5
**Confidence:** 4

**Summary:**

This paper proposes SpaceServe, a novel serving system that improves the efficiency of multimodal large language model (MLLM) inference by exploiting the complementary resource profiles of modality-specific encoders and the shared text decoder. The central idea is to shift from time-multiplexed scheduling—where encoders and decoders execute sequentially on the GPU—to space-multiplexed execution, in which both stages run concurrently on the same device through fine-grained streaming multiprocessor (SM) partitioning. To enable this, the authors introduce two key components: (1) a Space-Inference Scheduler (SIS) that dynamically allocates GPU resources using a cost model derived from operator-level profiling, and (2) a Time-Windowed Shortest Remaining Time First (TWSRFT) encoder scheduler that minimizes latency and smooths request arrivals. Through comprehensive empirical evaluation, SpaceServe is shown to deliver substantial speedups—up to 28.9× in time-per-output-token (TPOT)—across multiple model scales and request rates.

**Questions:**

See above.

**Ethical Concerns:**

["NO or VERY MINOR ethics concerns only"]

**Final Justification:**

Thank you for the detailed rebuttal. The response addresses my concerns on sparse architectures and generalizability, and I appreciate the additional clarifications and experiments provided.

**Limitations:**

Yes. The authors acknowledge the primary limitation regarding Time To First Token performance.

**Quality:**

4

**Strengths And Weaknesses:**

The paper addresses a real and increasingly salient challenge in multimodal LLM serving: how to make efficient use of heterogeneous GPU workloads when modality-specific encoders and a shared autoregressive decoder exhibit asymmetric compute and memory demands. The authors build a compelling case for space-multiplexing, backed by careful arithmetic intensity analysis and a cost-aware design that balances encoder latency against decoder throughput. Their implementation demonstrates strong engineering merit and outperforms the current state-of-the-art (vLLM) by a wide margin, particularly in high-load scenarios. Conceptually, the disaggregation and co-location strategy is not entirely new—earlier work on prefill/decode disaggregation and chunked execution exists—but this paper is one of the first to apply these ideas to the multimodal setting with fine-grained intra-GPU partitioning. This added granularity, along with the encoder-specific scheduler, constitutes a meaningful advancement.

Several aspects warrant careful consideration.

(major) The relationship between SpaceServe's serving-time optimizations and recent developments in multimodal architecture design merits discussion. Specifically, the work could benefit from consideration of how spatial multiplexing strategies might interact with sparse architectural approaches, such as those explored in "Mixture-of-Transformers: A Sparse and Scalable Architecture for Multi-Modal Foundation Models" (TMLR 2025). While SpaceServe optimizes resource allocation for existing dense models, sparse architectures inherently reduce computational requirements through structural design choices. Understanding whether the benefits of spatial multiplexing would be preserved or modified when applied to inherently sparse multimodal architectures could provide valuable insights for future system design considerations.


(minor) The resource utility curve construction relies on offline profiling, which may not fully capture dynamic workload variations encountered in production environments. The evaluation scope, while comprehensive within its domain, focuses primarily on Qwen family models, potentially limiting the generalizability of findings to other multimodal architectures with different resource characteristics. The cost model's adaptability to emerging architectural patterns remains unclear, particularly as the field continues to evolve toward more diverse encoder designs and cross-modal interaction mechanisms.

---

> ### Author Rebuttal · Authors · 2025-07-30
>
> Dear Reviewer  aTid,
>
> Thank you for your exceptionally thorough and insightful review. We are truly encouraged by your "excellent" ratings and your deep appreciation for SpaceServe's core contributions. Your summary accurately captures the essence of SpaceServe, and we are grateful for your recognition of its novelty and engineering merit.
>
> Your forward-looking questions, particularly regarding sparse architectures, are highly stimulating and have prompted us to think more deeply about the future trajectory of our work.Below, we address your specific concerns in detail:
>
> ### 1\. (Major) On the Relationship with Sparse Architectures
>
> We greatly appreciate your insightful question about the interplay between SpaceServe and sparse architectures, such as the Mixture-of-Transformers (MoT) referenced in your review. We agree that exploring the synergy between system-level optimization and architectural innovations is critical. To address this, we have added a detailed discussion in the revised paper.
>
>
> Our position is that **SpaceServe's spatial multiplexing and architectural approaches like MoT are not only compatible but are in fact also deeply complementary, yielding significant synergistic benefits.**
>
>
>
> **A. Theoretical Synergy with Mixture-of-Transformers (MoT):**
>
> The MoT architecture provides compelling architectural-level evidence for our system's core premise. By creating modality-specific parameters (e.g., separate FFNs for image vs. text), MoT validates that different modalities have distinct computational profiles that benefit from specialized processing.
>
> However, this very design introduces a new, fine-grained scheduling challenge during inference. When serving an MoT model, only one modality's "path" (e.g., the image FFN) is active at any given time for a single request, leaving other modality-specific components idle. This resource under-utilization is precisely the challenge SpaceServe is designed to solve. Our spatial scheduler can dynamically co-locate the execution of one request's compute-intensive image path with another request's memory-intensive text path on the same GPU. This maximizes hardware utilization and effectively translates MoT’s architectural efficiency into enhanced serving throughput.
>
>
> **B. Empirical Evidence with Modern MoE-based MLLMs:**
>
> Sparse architectures can be found in the latest models. Many new MLLMs, such as **DeepSeek-VL-v2** and **Kimi-VL**, adopt a hybrid "Vision Encoder \+ MoE-LLM" structure. To empirically validate our system's effectiveness on such architectures, we conducted new experiments on these models under a high-concurrency setting (10 requests/second).
>
>
> The results are conclusive:
>
> | Model (MoE-based) | Framework | TTFT | TPOT (ms) | TPOT Speedup |
> | :--- | :--- | :--- | :--- | :--- |
> | **DeepSeek-VL-v2** | vLLM | 5597ms | 122.45 | baseline |
> | | **SpaceServe (Ours)** | 4495ms | 30 | **4.08x** |
> | **Kimi-VL** | vLLM | 57.9s | 482 | baseline |
> | | **SpaceServe (Ours)** | 56.3s | 49 | **9.84x** |
>
>
> This "dense-sparse" hybrid architecture exhibits a pronounced resource heterogeneity: a dense, compute-intensive vision encoder followed by a sparse, memory-intensive MoE decoder. Our **4x-9.8x speedup** on these models provides powerful evidence that SpaceServe effectively manages this new form of workload asymmetry.
>
> In summary, whether the sparsity is achieved by "modality-specific paths" (like MoT) or by "component-specific sparsity" (like MoE-LLMs), the fundamental result is increased resource heterogeneity. Our work provides a general, system-level solution to harness this heterogeneity, demonstrating SpaceServe's lasting value for efficiently serving the next generation of MLLMs.
>
> ### 2\. (Minor) On Generalizability and Offline Profiling
> We appreciate your points regarding model generalizability and the limitations of offline profiling, which are critical for ensuring SpaceServe’s broad applicability and practicality.
>
> * **Model Generalizability:** As noted above, our new experiments now include a wide range of architectures (LLaVA, MiniCPM-V, and the MoE-based DeepSeek-VL and Kimi-VL). The results, summarized below, confirm SpaceServe’s robust performance across varied architectures:
>
>
> | Model | Framework | TTFT (ms) | TPOT (ms) | TPOT Speedup |
> | :---- | :---- | :---- | :---- | :---- |
> | **MiniCPM-V2.6** | vLLM | 1876 | 100.11 | baseline |
> |  | **SpaceServe (Ours)** | **1486** | **40.80** | **2.45x** |
> | **DeepSeek-VL-v2** | vLLM | 5597 | 122.45 | baseline |
> |  | **SpaceServe (Ours)** | **4495** | **30** | **4.08x** |
> | **Kimi-VL** | vLLM | 57900 | 482 | baseline |
> |  | **SpaceServe (Ours)** | **56323** | **49** | **9.84x** |
> | **LLaVA-1.6** | vLLM | 20351 | 102 | baseline |
> |  | **SpaceServe (Ours)** | 19885 | 44 | **2.32x** |
>
>
> * **Offline Profiling:** You raised a valid point about the limitations of offline profiling. Our choice was a pragmatic one, trading dynamic adaptation for the stability and zero-overhead benefits that are standard in high-performance systems. Importantly, our framework is modular. It can adapt to new architectures, including the specific paths of an MoT model, by simply running a new profiling script to generate an updated cost profile without changing the core scheduling logic.
> ---
> Thank you again for your insightful review in supporting our work. Your comments have strengthened our paper by connecting SpaceServe to cutting-edge architectural advancements, particularly sparse (sparsity and MoT), and by prompting us to clarify our approach to generalizability and profiling. We hope our revision, including new experimental data and detailed discussion, meet your high standards and fully address you concerns.

---

> ### Comment · Area_Chair_Niag · 2025-08-04
> **Dear Reviewer Please Engage in Discussion (2nd Reminder)**
>
> Dear Reviewer,
>
> The authors have replied to your concerns, and it would be very helpful if you could engage in a discussion to clarify your concerns.
>
> Best,
>
> AC

---

> > ### Comment · Area_Chair_Niag · 2025-08-05
> > **Reminder about Engaging in Discussion (Deadline Aug. 8th)**
> >
> > Dear Reviewer,
> >
> > This is a friendly reminder about engaging in the discussion.
> >
> > Best,
> >
> > AC

---

> > > ### Comment · Area_Chair_Niag · 2025-08-08
> > >
> > > Dear Reviewer aTid,
> > >
> > > Although you submitted your mandatory acknowledgement, you did not engage in the discussion. Note that this can be grounds for flagging this as insufficient review which may incur penalties. Please engage in the discussion since today is the last day to do so.
> > >
> > > Best,
> > >
> > > AC

---

### Official Review · Reviewer_tKdj · 2025-07-04

**Clarity:** 2
**Significance:** 3
**Originality:** 3
**Rating:** 5
**Confidence:** 2

**Summary:**

This paper introduces SpaceServe, a method to more efficiently serve multimodal LLMs.
SpaceServe takes advantage of compute/memory tradeoffs between encoders and decoder with spatial multiplexing instead of time multiplexing. SpaceServe also proposes a scheduler to order encoder requests.
Empirical results show that the proposed method is more efficient than the vLLM baseline.

**Questions:**

* Does this generalize to architectures with drastically different shapes and to MLLM with multiple modalities on the output?
* Are there other baselines than vllm? If so, how does this work compare to those baselines?
* Given that vllm is the go to library for serving LLMs, could this approach be integrated into vllm? I did not see any mention of releasing the method open source. Is there such a plan?
* If I understood correctly, the experiments are conducted on a limited set of GPUs. Could there be a sanity check that the findings hold at scale, i.e. with more gpus to parallelize the requests but also larger number of requests?
273: unaffected TTFT: TTFT is not reported in Figure 4

presentation comments:

I would suggest that the abstract mention 4.81x on average as well.
62: Mutliplexing → Multiplexing
74: complementary → complementarity
111: streaming multiprocessors. SM is used before. The abbreviation should be introduced once on first occurrence.
199: and performs preprocessing of multimodal data Request → and performs preprocessing of multimodal data. Request (missing period)

**Ethical Concerns:**

["NO or VERY MINOR ethics concerns only"]

**Final Justification:**

The rebuttal provides thorough answers + experiments, which alleviates most of my concerns so I'm raising the score from 4 to 5.

**Limitations:**

yes

**Quality:**

3

**Strengths And Weaknesses:**

Strengths:
* the community is increasingly focused on multimodal models. As a result, this work is critical for more efficient serving of such workloads
* the proposed method is effective in reducing time per output token (up to 28.9x)
* ablations are conducted on both spatial multiplexing and scheduler to demonstrate how each component of the proposed method contributes to efficiency

Weaknesses:
In general, weaknesses relate to more details/comments below. To highlight the main one:
* the method provides efficiency improvements but the tradeoff is complexity. This potential tradeoff could be discussed in detail. If there is a plan to integrate this method into vllm, this could go a long way in encouraging adoption for example.

---

> ### Author Rebuttal · Authors · 2025-07-30
>
> Dear Reviewer tKdj, Thank you for your thorough review and for recognizing the importance and effectiveness of SpaceServe. Your insightful questions regarding complexity, generalizability, and community adoption are critical, and we have incorporated them into our revisions to strengthen the paper.
>
> Below, we address each of your concerns in detail:
>
> ### 1\. On the Tradeoff between Efficiency and Complexity & Open-Sourcing
>
>
> We acknowledge the reviewer’s concern about the complexity introduced by SpaceServe and the need to justify it. While SpaceServe incorporates new scheduling logic to achieve its performance gains, this complexity is encapsulated within the framework. It remains transparent to end-users and does not impact usability. To facilitate community adoption and transparency, we are committed to **open-sourcing the complete code for SpaceServe.** We have added a statement to the conclusion of our revised paper to reflect this commitment. Furthermore, we recognize the value of integrating SpaceServe's principles into vLLM. Our open-source release will be structured to encourage such integration, and we are open to collaborating with the community to make this happen.
>
> ### 2\. On Generalization to Different Architectures
>
> We appreciate the reviewer’s critical question regarding whether the generalizability to MLLMs with "drastically different shapes" and multiple output modalities.
>
> First, to demonstrate the generalizability of our method, we conducted extensive new experiments across a range of diverse MLLMs with different architectures, including **LLaVA-1.6, MiniCPM-V2.6, DeepSeek-VL-v2, and Kimi-VL,** under a high-concurrency setting (10 requests/sec). The results, detailed below, confirm SpaceServe’s robust performance across varied architectures:
>
> | Model | Framework | TTFT (ms) | TPOT (ms) | TPOT Speedup |
> | :---- | :---- | :---- | :---- | :---- |
> | **MiniCPM-V2.6** | vLLM | 1876 | 100.11 | baseline |
> |  | **SpaceServe (Ours)** | **1486** | **40.80** | **2.45x** |
> | **DeepSeek-VL-v2** | vLLM | 5597 | 122.45 | baseline |
> |  | **SpaceServe (Ours)** | **4495** | **30** | **4.08x** |
> | **Kimi-VL** | vLLM | 57900 | 482 | baseline |
> |  | **SpaceServe (Ours)** | **56323** | **49** | **9.84x** |
> | **LLaVA-1.6** | vLLM | 20351 | 102 | baseline |
> |  | **SpaceServe (Ours)** | 19885 | 44 | **2.32x** |
>
> Second, regarding MLLMs with multiple output modalities (e.g., text and image generation), SpaceServe remains highly effective . The core of SpaceServe’s approach is to exploit resource heterogeneity. In MLLMs, the root of this heterogeneity lies in the contrast between the **compute-intensive vision encoding** and the **memory-intensive text decoding**. As long as a mode exhibits this resource asymmetry between its constituent tasks, regardless of output modalities, SpaceServe can leverage spatial multiplexing to optimize scheduling and improve throughput.
>  We have added a discussion to the revised paper to clarify this applicability to multimodal output models.
>
>
>
> ### 3\. On Other Baselines
>
> We chose vLLM as our primary baseline because it represents the state-of-the-art for high-throughput MLLM serving.
>
> We also considered other promising systems like SGLang. However, at the time we conducted our experiments, SGLang's support for MLLMs was still maturing and **lacked several important features necessary for robustly serving modern multimodal models.** Specifically, when serving newer models that support dynamic, high-resolution inputs (such as Kimi-VL, Qwen-VL, and DeepSeek-VL) under **high concurrency**, we found that this lack of feature support led to **out-of-memory (OOM) errors**, making a fair, apples-to-apples comparison under realistic load infeasible.
>
> In contrast, **vLLM has a larger community and offers more comprehensive and stable support for these demanding workloads**, making it the most suitable and robust baseline for our experiments. We will clarify this justification in the related work section.
>
> ### 4\. On Scalability (More GPUs & Requests)
>
> This is a great question. We have addressed it in two ways:
>
> * **Larger Number of Requests:** As mentioned, all our experiments were conducted under a **high load of 10 requests per second**. Our results, which show increasing benefits under load, directly confirm that our findings hold for a large number of requests.
>
> * **Scaling to More GPUs:** SpaceServe's intra-GPU optimization is **orthogonal and complementary** to inter-GPU scaling strategies. By making *each individual GPU* in a distributed system more efficient, SpaceServe amplifies the overall system throughput.
>
> ### 5\.  Clarification on TTFT Results
> Due to space constraints, the full Time to First Token (TTFT) results were placed in Appendix B. For your convenience, we present the key TTFT data below, which shows that SpaceServe's TTFT is on par with vLLM.
>
> | Model | Method | TTFT (s) @ 2 RPS | @ 4 RPS | @ 6 RPS | @ 8 RPS | @ 10 RPS |
> | :--- | :--- | :--- | :--- | :--- | :--- | :--- |
> | **Qwen-2-VL-2B** | vLLM | 12.068 | 16.180 | 18.655 | 23.035 | 26.038 |
> | | **SpaceServe** | 13.324 | 17.515 | 21.114 | 24.726 | 27.437 |
> | **Qwen-2-VL-7B** | vLLM | 19.492 | 22.033 | 30.306 | 34.745 | 38.795 |
> | | **SpaceServe** | 18.832 | 22.419 | 32.050 | 36.533 | 39.086 |
> | **Qwen-2.5-VL-32B** | vLLM | 16.553 | 32.631 | 41.849 | 46.440 | 47.547 |
> | | **SpaceServe** | 17.131 | 33.242 | 42.489 | 47.144 | 48.966 |
> | **Qwen-2-VL-72B** | vLLM | 14.872 | 26.647 | 35.717 | 39.780 | 42.067 |
> | | **SpaceServe** | 14.810 | 26.633 | 36.321 | 38.378 | 43.834 |
> **Time To First Token (TTFT) comparison (unit: seconds) across different models with varying Request Per Second (RPS). Lower TTFT is better.**
>
>
> ### 6\. On Presentation Comments
>
> Thank you for these meticulous suggestions\! We have made all the corrections you pointed out in the revised manuscript, including adding the average speedup to the abstract, fixing typos, and ensuring TTFT results are clearly reported in Figure 4 and the text.
>
> ---
>
> Once again, we thank you for your insightful feedback, which has helped us significantly improve the clarity, scope, and impact of our paper. We hope that our responses and the extensive new experimental data have addressed your concerns.

---

> > ### Comment · Reviewer_tKdj · 2025-08-04
> >
> > Thanks a lot for the thorough response. This alleviates most of my concerns.
> >
> > > Second, regarding MLLMs with multiple output modalities (e.g., text and image generation), SpaceServe remains highly effective.
> >
> > Do you have experiments on this?
> >
> > Best,
> >
> > -Reviewer

---

> ### Author Response · Authors · 2025-08-04
> **Response to Reviewer tKdj**
>
> Dear Reviewer,
>
> Thank you for your insightful follow-up question. This is an excellent and forward-looking point, and we appreciate the opportunity to clarify how SpaceServe’s principles apply to MLLMs with multimodal output capabilities, such as text-to-image generation.
>
> Theoretically, these advanced models are well-suited for SpaceServe's spatial multiplexing approach. Their architectures inherently create a processing pipeline with highly heterogeneous resource demands, often involving three distinct stages:
>
> 1. A **compute-intensive** vision encoder for input processing. Taking DeepSeek Janus Pro as a case study, its vision encoder processes a fixed 384x384 input image. The Arithmetic Intensity for this stage is approximately 317 FLOPS/Byte, characterizing it as a classic compute-bound task.
>    $$
>    \text{Arithmetic Intensity} = \frac{332.4 \text{B FLOPS}}{1.05 \text{GB}} \approx 317 \text{ FLOPS/Byte}
>    $$
> 2. A **memory-bound** autoregressive LLM for processing. During the autoregressive decoding phase, the workload transitions dramatically. The Arithmetic Intensity for the Janus-pro's language model is extremely low.
>    $$
>    \text{Arithmetic Intensity} \approx 1 \text{ FLOPS/Byte}
>    $$
> 3. A second **compute-intensive** stage for generating the visual output. In the case of DeepSeek Janus Pro, its VQ-based image decoder once again exhibits high computational demands, reverting to a compute-intensive profile.
>    $$
>    \text{Arithmetic Intensity} = \frac{90 \text{B FLOPS}}{240 \text{MB}} = 375 \text{ FLOPS/Byte}
>    $$
> It creates an ideal scenario for co-locating the memory-hungry LLM kernel with the compute-heavy vision kernels to maximize GPU utilization and overall throughput.
>
> Regarding empirical data, the reason we have not yet presented experiments on these models is a practical one. Our current implementation of SpaceServe is built upon vLLM, a mainstream serving framework. At present, vLLM does not offer support for the serious models(e.g., Deepseek Janus-pro and ILLUME+).
>
> We are confident in the applicability of our work to these models and plan to provide a full experimental evaluation as soon as frameworks like vLLM extend their support to these architectures in the future.
>
> Thank you again for your valuable engagement.
>
> Best regards,
>
> The Authors

---

> > ### Comment · Area_Chair_Niag · 2025-08-08
> >
> > Dear Reviewer tKdj,
> >
> > The authors have replied to your comments. Please read it and make sure it addresses your concerns.
> >
> > Best,
> >
> > AC

---

### Official Review · Reviewer_Sa3P · 2025-07-05

**Clarity:** 3
**Significance:** 3
**Originality:** 3
**Rating:** 5
**Confidence:** 2

**Summary:**

This paper proposes a novel SpaceServe framework for efficient inference on MLLMs. The key insight is that the resource asymmetry of small encoders and large decoders significantly limits the efficiency of existing frameworks. SpaceServe proposes a space-multiplexing strategy to parallelize the encoding and decoding processes and fully explore the potential of the hardware. Experiments on the Qwen-VL series models show that SpaceServe largely reduced the TPOT.

**Questions:**

1. Efficiency of different frameworks on different input resolutions would be helpful to have a more thorough understanding.

**Ethical Concerns:**

["NO or VERY MINOR ethics concerns only"]

**Final Justification:**

My concerns on the limitation of model tested and experimental setting are solved. The author's discussion on hardware and platform is reasonable and I raise my score to 5.

**Limitations:**

yes

**Quality:**

3

**Strengths And Weaknesses:**

Strengths

1) The motivation and insight are sound, as the encoding and decoding process efficiency bottleneck discrepancy. The encoding process is computation-intensive while the decoding process is more memory-intensive.
2) The proposed SpaceServe method exhibits high performance compared with the vLLM baseline by x3 to x10 times.
3) The proposed method could be helpful for the application of MLLMs and further research in this field.

Weakness

1) The tested models are limited to only the Qwen-VL series, while omitting other MLLMs with different architectures, such as LLaVA and MiniCPM-V.
2) The tested hardware configuration and platforms are limited.

---

> ### Author Rebuttal · Authors · 2025-07-30
>
> Dear Reviewer Sa3P,
> We appreciate your thorough and constructive feedback on our paper.
> We are pleased that the motivation, insights, and performance benefits of SpaceServe were well-received. Your comments have been invaluable in refining our work.
>
> In response to your questions, we have conducted additional experiments and revised the manuscript. These updates resolve the raised concern and strengthen the paper’s contribution.
>
>
> ### 1\. On the Weakness of Limited Model Diversity (Generalizability of SpaceServe)
>
>
> We agree that demonstrating generalizability is essential, and thank you for raising this point. Below, we clarify our initial choice of Qwen-VL series and present new experiments showcasing SpaceServe’s broad applicability.
>
> **Rationale for Qwen-VL Selection**
>
>
> We selected the Qwen-VL series because of its top popularity in MLLM benchmarks at the time of our experiment. Its advanced features, such as dynamic resolution and long context support, make it a robust testbed for SpaceServe. Significant performance gains on such a demanding model validate our approach’s effectiveness.
>
> **Expanded Evaluation**
>
> To address concerns about broader applicability, we conducted new experiments across diverse model architectures under a high-concurrency workload (10 requests/second) to mirror real-world conditions.
>
> * **Established Architectures**
>   * LLaVA-1.6: This older MLLM with fixed-resolution architecture is less vision-intensive. SpaceServe achieved a 2.32x speedup.
>   * MiniCPM-V2.6: A model that is primarily optimized for edge devices. SpaceServe received 2.45x speedup.
>
> 	Those results show that our method delivers strong performance, even in models with reduced compute-memory heterogeneity between the encoder and the LLM.
>
> Those results demonstrate our method performs reasonably well even on models where the compute-memory heterogeneity between the encoder and the LLM is less pronounced.
>
> * **Modern MoE-based Architectures**
>   * DeepSeek-VL-v2 and Kimi-VL: These recent models with dynamic resolution and Mixture-of-Experts (MoE) layers are highly relevant and well-suited for optimization using our method. SpaceServe delivered impressive speedups of 4.08x and 9.84x, respectively, which prove its effectiveness and generalizability for diverse cutting-edge MLLM architectures.
> The results, summarized below, demonstrate a clear and consistent advantage for SpaceServe across all tested architectures:
>
> | Model | Framework | TTFT (ms) | TPOT (ms) | TPOT Speedup |
> | :---- | :---- | :---- | :---- | :---- |
> | **LLaVA-1.6** | vLLM | 20351 | 102 | baseline |
> |  | **SpaceServe (Ours)** | 19885 | 44 | **2.32x** |
> | **MiniCPM-V2.6** | vLLM | 1876 | 100.11 | baseline |
> |  | **SpaceServe (Ours)** | 1486 | 40.80 | **2.45x** |
> | **DeepSeek-VL-v2** | vLLM | 5597 | 122.45 | baseline |
> |  | **SpaceServe (Ours)** | 4495 | 30 | **4.08x** |
> | **Kimi-VL** | vLLM | 57900 | 482 | baseline |
> |  | **SpaceServe (Ours)** | 56323 | 49 | **9.84x** |
>
> The particularly high speedup on Kimi-VL (9.84x) is noteworthy and highlights a key strength of our system. Kimi-VL's ability to support exceptionally high-resolution inputs, while enabling superior multimodal understanding, also drastically increases the computational intensity of its vision encoder. This creates a more pronounced resource heterogeneity between the compute-intensive encoder and the memory-intensive LLM decoder—a scenario where SpaceServe's spatial multiplexing provides the greatest benefit. We observe that this architectural trend is becoming a standard for state-of-the-art models, as seen in other recent releases like Seed-VL and Keye-VL. This underscores the growing importance of our approach for efficiently serving the next generation of powerful MLLMs.
>
> ### 2\. On the Question of Efficiency with Different Input Resolutions (Efficiency Across Different Input Resolutions)
>
>
> We appreciate the reviewer’s insightful question regarding SpaceServe’s performance under varying input resolutions. It is critical to understand its efficiency. To address this, we conducted new experiments evaluating SpaceServe across a range of input resolutions, from `224x224` to `2048*2048`, under a high-concurrency workload (10 requests per second).
>
> The results exceeded our expectations and underscore a key strength of our approach:
>
> | Resolution | Framework | TTFT | TPOT (ms) | TPOT Speedup |
> | :---- | :---- | :---- | :---- | :---- |
> | **224 \* 224** | vLLM | 90.63ms | 23.62 | baseline |
> |  | **SpaceServe (Ours)** | 93.00ms | **17.23** | **1.37x** |
> | **512 \* 512** | vLLM | 210.50ms | 38.59 | baseline |
> |  | **SpaceServe (Ours)** | 205.42ms | **25.85** | **1.49x** |
> | **1k \* 1k** | vLLM | 15.31s | 212.58 | baseline |
> |  | **SpaceServe (Ours)** | 15.54s | **35.59** | **5.97x** |
> | **2k \* 2k** | vLLM | 139.35s | 470.66 | baseline |
> |  | **SpaceServe (Ours)** | 135.76s | **37.99** | **12.39x** |
>
> In our experiments, we observe that as image resolution increases, the vision encoder’s computational demand significantly degrades vLLM’s performance in terms of increasing both TTFT and TPOT. In contrast, SpaceServe maintains a remarkably low and stable TPOT across all resolutions.
>
> Notable, SpaceServe’s TPOT speedup over vLLM scales dramatically with resolution, from 1.37x at 224x224 to an impressive 12.39x at 2048x2048. This demonstrates that our spatial multiplexing approach effectively mitigates the latency of compute-intensive tasks. Far from being a limitation, high-resolution inputs highlight SpaceServe’s great strength. These important findings and corresponding analysis are now a core component of our revised paper.
>
> ### 3\. On the Weakness of Limited Hardware Configurations
>
>
> We acknowledge the reviewer’s point that evaluating SpaceServe on diverse hardware would further validate our claim. To clarify, SpaceServe’s design leverages fundamental features of modern GPUs, which are the separation of streaming multiprocessors (SMs) and memory controllers to co-locate compute-intensive (encoder) and memory-intensive (decoder) tasks. This architectural principle is ubiquitous across modern NVIDIA GPUs
>
> Due to time and hardware constraints during this rebuttal period, we were unable to conduct a comprehensive hardware evaluation. We have added this in the "Limitations and Future Work" section of our revised paper and highlighted diverse hardware testing as a key direction for future research
>
> ---
>
>
> In summary, thanks to your insightful feedback, we have significantly strengthened our paper. The new experiments provide robust evidence of SpaceServe’'s **generalizability across diverse MLLM architectures** and its **superior performance under heavy computational loads, particularly with high-resolution images**.
>
>
> We sincerely thank you for your time and valuable guidance. We hope our revisions and detailed responses have fully addressed your concerns.

---

> ### Comment · Area_Chair_Niag · 2025-08-04
> **Dear Reviewer Please Engage in Discussion (2nd Reminder)**
>
> Dear Reviewer,
>
> The authors have replied to your concerns, and it would be very helpful if you could engage in a discussion to clarify your concerns.
>
> Best,
>
> AC

---

> > ### Comment · Area_Chair_Niag · 2025-08-05
> > **Reminder about Engaging in Discussion (Deadline Aug. 8th)**
> >
> > Dear Reviewer,
> >
> > This is a friendly reminder about engaging in the discussion.
> >
> > Best,
> >
> > AC

---

> > > ### Comment · Area_Chair_Niag · 2025-08-08
> > >
> > > Dear Reviewer Sa3P,
> > >
> > > The authors provided an answer and please engage at least today. Note that this can be ground for flagging the review and interaction as insufficient, which may incur penalties.
> > >
> > > Best,
> > >
> > > AC

---

### Note · Authors · 2025-08-13

Dear Reviewers (Sa3P, tKdj, aTid) and Area Chair (Niag),

As the discussion period concludes, we would like to express our sincerest gratitude for your invaluable engagement with our work on SpaceServe. Your thoughtful critiques, insightful questions, and forward-looking suggestions have been instrumental in helping us refine and significantly strengthen our paper.

We are delighted that the core motivation and contributions of SpaceServe—addressing the critical resource heterogeneity in MLLMs through spatial multiplexing—were well-received. Throughout this process, your feedback has guided us in generating substantial new experimental evidence that we believe has rigorously addressed the key concerns raised.

To summarize the enhancements made thanks to your guidance:

1.  **Broadened Generalizability:** We have moved beyond the Qwen-VL series and demonstrated SpaceServe's robust performance across a diverse set of MLLM architectures, including established models like **LLaVA-1.6** and **MiniCPM-V2.6**, and cutting-edge, sparse **MoE-based models** like **DeepSeek-VL-v2** and **Kimi-VL**. The consistent and significant speedups (ranging from **2.32x** to **9.84x**) provide strong evidence of our method's wide applicability.

2.  **Validated Performance under Varying Resolutions:** We conducted new experiments showing that SpaceServe's performance advantage not only holds but **dramatically scales with increasing input resolution**. The TPOT speedup escalating from **1.37x** at 224x224 to **12.39x** at 2Kx2K highlights that our approach excels precisely where modern MLLMs are headed: processing high-fidelity, compute-intensive visual inputs.

3.  **Clarified Future Architectural Synergy:** We have added a detailed discussion on the profound and complementary relationship between SpaceServe's system-level optimizations and architectural innovations like **Mixture-of-Experts (MoE)** and **Mixture-of-Transformers (MoT)**.

We have diligently incorporated these new results and discussions into the revised manuscript. We believe the paper is now much stronger, offering a more comprehensive validation of SpaceServe's effectiveness, generalizability, and relevance to the future of multimodal AI serving.

Thank you once again for your time, expertise, and constructive feedback. Your efforts have pushed us to improve our work in meaningful ways, and for that, we are truly grateful.

Sincerely,

The Authors of Submission #11944

---

### Decision · Program_Chairs · 2025-09-17

**Decision:**

Accept (poster)

**Comment:**

This paper presents SpaceServe, a serving system for multimodal large language models that improves GPU utilization by space-multiplexing modality-specific encoders and the shared text decoder. It introduces a Space-Inference Scheduler (SIS) and a Time-Windowed Shortest-Remaining-First (TWSRFT) policy to optimize SM partitioning and request batching. Experiments on Nvidia A100 GPUs show that SpaceServe achieves up to 29.71× speedup over existing serving stacks.

The concerns raised in the reviews and that were discussed can be summarized as follows:
1. Insufficient baselines (i.e., adding other VLMs besides Qwen family) in the experiments and limited hardware variations (R-Sa3P, R-aTid);
2. Lacking discussion about efficiency vs complexity tradeoff (R-tKdj); and
3. Lacking discussion about the relationship of SpaceServe and sparse architectures (R-aTid).

During the discussion between reviewers and authors, the authors provided new experiments and thorough clarifications that satisfied all reviewers. Given this positive discussion, reviewers recommended to accept the paper unanimously. Thus, we encourage the authors to include any additional clarification and experiments that help the story of the paper.